# Levels and predictors of anxiety, depression, and stress during COVID-19 pandemic among frontline healthcare providers in Gurage zonal public hospitals, Southwest Ethiopia, 2020: A multicenter cross-sectional study

**Fisha Alebel GebreEyesus**[1]*, **Tadesse Tsehay Tarekegn**[1], **Baye Tsegaye Amlak**[1], **Bisrat Zeleke Shiferaw**[1], **Mamo Solomon Emeria**[1], **Omega Tolessa Geleta**[1], **Tamene Fetene Terefe**[1], **Mtiku Mammo Tadereregew**[2], **Melkamu Senbeta Jimma**[3], **Fatuma Seid Degu**[1,4], **Elias Nigusu Abdisa**[5], **Menen Amare Eshetu**[6], **Natnael Moges Misganaw**[7], **Ermias Sisay Chanie**[7]

1 Department of Nursing, College of Medicine and Health Sciences, Wolkite University, Wolkite, Ethiopia, 2 Department of Biomedical Sciences, College of Medicine and Health Sciences, Wolkite University, Wolkite, Ethiopia, 3 Department of Nursing, College of Health Sciences, Assosa University, Assosa, Ethiopia, 4 Department of Nursing, College of Health Sciences, Wollo University, Wollo, Ethiopia, 5 Department of Nursing, College of Medical and Health Science, Wachemo University, Hossana, Ethiopia, 6 Department of Nursing, College of Medicine and Health Science, Mizan Tepi University, Mizan Tepi, Ethiopia, 7 Department of Pediatric and Child Health Nursing, College of Health Sciences, Debre Tabor University, Debre Tabor, Ethiopia

* fishalebel@gmail.com

## Abstract

### Introduction

The provision of quality health care during the COVID-19 pandemic depends largely on the health of health care providers. However, healthcare providers as the frontline caregivers dealing with infected patients, are more vulnerable to mental health problems. Despite this fact, there is scarce information regarding the mental health impact of COVID-19 among frontline health care providers in South-West Ethiopia.

### Objective

This study aimed to determine the levels and predictors of anxiety, depression, and stress during the COVID-19 pandemic among frontline healthcare providers in Gurage zonal public hospitals, Southwest Ethiopia, 2020.

### Methods

An institutional-based cross-sectional study was conducted among 322 health care providers from November 10–25, 2020 in Gurage zonal health institutions. A simple random sampling technique was used to select the study participants. A pretested self-administered structured questionnaire was used as a data collection technique. The data were entered into the Epi-data version 3.01 and exported to SPSS version 25.0 for analysis. Both

**Data Availability Statement:** All relevant data are within the manuscript and its Supporting Information files.

**Funding:** The author(s) received no specific funding for this work.

**Competing interests:** The authors have declared that no competing interests exist.

**Abbreviations:** AOR, Adjusted Odds Ratio; CoV, Corona Virus; COVID-19, Corona Virus Disease 2019; CSA, Central Statistics Agency; GAD, Generalized Anxiety Disorder; HC, Health Center; HCPs, Health Care Providers; HCWs, Health Care Workers; ICN, International Council of Nurses; IPC, Infection Prevention and Control; LMICs, Low and Middle Income Countries; NGO, Non-Governmental Organization; NICU, Neonatal Intensive Care Unit; OPD, Out Patient Department; OR, Operation Room; PHEIC, Public Health Emergency of International Concern; PHQ, Patient Health Questionnaire; PPE, Personal Protective Equipment's; PSS, Perceived Stress Scale; PTSD, Post Traumatic Distress Syndrome; SARS-CoV-2, Severe Acute Respiratory Syndrome Coronavirus 2; SD, Standard Deviation; SNNPRE, South Nation Nationality and Peoples Regions of Ethiopia; SPSS, Statistical Package for Social Sciences; WHO, World Health Organization.

descriptive statistics and inferential statistics (chi-square tests) were presented Bivariable and Multivariable logistic regression analyses were made to identify variables having a significant association with the dependent variables.

## Results

The results of this study had shown that the overall prevalence of anxiety, depression and stress among health care providers during the COVID-19 pandemic was 36%, [95% CI = (30.7%- 41.3%)], 25.8% [95% CI = (21.1%- 30.4%)] and 31.4% [95% CI = (26.4%- 36.0%)] respectively. Age, Adjusted Odds Ratio [AOR = 7.9], Educational status, [AOR = 3.2], low monthly income [AOR = 1.87], and presence of infected family members [AOR = 3.3] were statistically associated with anxiety. Besides this, gender, [AOR = 1.9], masters [AOR = 10.8], and degree holder [AOR = 2.2], living with spouse [AOR = 5.8], and family [AOR = 3.9], being pharmacists [AOR = 4.5], and physician [AOR = (0.19)], were found to be statistically significant predictors of depression among health care providers. Our study finding also showed that working at general [AOR = 4.8], and referral hospitals [AOR = 3.2], and low monthly income [AOR = 2.3] were found to be statistically significant predictors of stress among health care providers.

## Conclusion

Based on our finding significant numbers of healthcare providers were suffered from anxiety, depression, and stress during the COVID-19 outbreak. So, the Government and other stakeholders should be involved and closely work and monitor the mental wellbeing of health care providers.

## Introduction

Coronavirus (CoV) infections are emerging respiratory viruses that are known to cause illnesses ranging from the common cold to severe acute respiratory syndrome (SARS) [1]. As of January 30th, 2020, the "World Health Organization" (WHO) characterized the ongoing COVID-19 outbreak as a "Public Health Emergency of International Concern" (PHEIC) [2], and later, due to uncased fast spread, the severity of illness, the continual escalation in several affected countries, cases and causalities, WHO declared coronavirus disease 2019 (COVID-19) a global pandemic on 11 March 2020 [3].

The pandemic could have severe effects on the mental health of the general population and health care providers(HCPS) [4]. As a result, people have been comparing the emergence of a novel Coronavirus (2019-nCoV) to "the end of the world," and the whole world reacts to the event with panic, insomnia, stress, irritability, and feelings of distractibility [5].

Healthcare providers are always at the forefront in the response to emerging infectious disease outbreaks which are encountering many sources of stress, and recent evidence showed that the COVID-19 pandemics can undermine not only physical health but also take a toll on these providers' mental health and resilience [6, 7]. In a Chinese study, researchers found that a considerable proportion of participants reported symptoms of anxiety (44.6%), moderate to severe depression (50.4%), insomnia (34%), and moderate to severe psychological distress (71.5%) [8]. In addition to this, studies carried out in Italy revealed that 50.1% of participants

reported symptoms of clinically relevant anxiety, 26.6% symptoms of depression and 53.8% showed symptoms of post-traumatic distress [9].

Mental health and psychosocial consequences of the COVID-19 pandemic may be particularly serious for health professionals because HCPs often have to respond to demanding and unforeseen medical emergencies [10]. In the initial phase of the SARS-CoV-2 outbreak, 29% of all hospitalized patients were HCPs [11]. A recent report from the International Council of Nurses (ICN), found that health worker infections ranged from 1–32% of all confirmed COVID-19 cases [12].

Globally, there have been more than 230,920,739 infections and 4,733,350 fatalities after the declaration of the pandemic by the WHO. In Africa, there are about 8,269,298 confirmed cases and 207,760 deaths reported as of September 23 /2021 [13]. According to the Amnesty International report, 17,000 health workers have died worldwide from COVID-19 over the last year, which implied that one health care worker was dying every 30 minutes, which was a "tragedy and an injustice" [14].

Ethiopia is one of the countries threatened by COVID- 19, with a total of 336,762 confirmed cases and 5,254 registered deaths as of September 23/2021 [13]. It is now the leading country in East Africa with the highest number of infected people. Thousands of HCPs have been infected with COVID-19 [15]. To minimize the risk of COVID-19 transmission in the community, the Ethiopian government declared an emergency and mandated compulsory physical distancing, the establishment of isolation and quarantine centers for suspected and confirmed cases, the activation of the Federal Emergency Operation Center, frequent hand washing, temporary closure of schools and higher education institutions, establishing alternative working modalities for public servants, and the suspension of mass gatherings [16, 17]. Despite the ongoing preventative and control measures, containing the spread of the virus could be challenging in light of the underlying social and infrastructural settings of the country.

The global COVID-19 pandemic has created a massive public health crisis and several challenges for healthcare providers [6]. The social, economic, and health effects are extensive, where they are related to increased all-cause mortality, occupational disability, poor quality of life, and cardiovascular disease risk [18]. Despite its multiple consequences, mental health is often neglected as a public health agenda [19].

The psychological effects related to the current pandemic are caused by numerous factors, including competency concerns when redeployed without adequate training, uncertainty about the duration of the crisis, misleading information about the effectiveness of the vaccine, depletion of personal protection equipment, feelings of being inadequately supported, the hefty workload, the need to take stressful precautions during the medical examination/ in the operative fields and frequent exposure to patients' suffering and dying [10, 20–24].

Studies also showed that those health care workers who feared contagion and infection of their family, friends, and colleagues felt uncertainty and stigmatization [25, 26], reported reluctance to work or contemplate resignation and reported experiencing high levels of stress, anxiety, and depression symptoms which could have long term psychological implications. Similar concerns about the mental health, psychological adjustment, and recovery of health care workers treating and caring for patients with COVID-19 are now arising [25, 27].

To decrease the extent of the psychological consequences, some measures are taken such as avoiding intense exposure to COVID-19 media coverage, providing resilience training for HCPs, maintaining a compassionate and positive lifestyle by providing support to others [28]. Besides this, WHO called for action to address the immediate needs and measures needed to save lives and prevent a serious impact on the physical and mental health of healthcare providers [29].

A number of research articles published over the past few months showed that a significant proportion of healthcare providers who worked within primary, secondary, and tertiary hospitals developed adverse mental outcomes while providing service for the needy population [30–34]. Despite this fact, sufficient information is not available regarding the mental health impact of COVID-19 among frontline health care providers in South-West Ethiopia. So, the current study aimed to determine the levels and determinants of anxiety, depression, and stress among frontline healthcare providers in Gurage zonal public hospitals.

## Methods and materials

### Study design

An institutional-based cross-sectional study design was conducted.

### Study period and area

The study was conducted in the Gurage zonal public health institutions of SNNPRE from October–December / 2020. Gurage Zone is one of the fifteen zones and four special woredas found in SNNPR state. Wolkite town is the capital of Gurage zone which is located 158 Km southwest of Addis Ababa and 260 Km from Hawasa. It has 20 woreda and two municipalities. According to the 2012 population projection by CSA the total population is 1,767,518.

There are seven hospitals in the Gurage zone. Five of the hospitals in the zone are primary hospitals, one general hospital and the remaining one is a specialized comprehensive hospital, there are 79 health centers (7 are NGO HC) and 444 Functional health posts serving the total population in the zone. There is also a COVID-19 testing center; some hospitals are readily organized to serve quarantine and treatment centers [35].

### Source populations

All health care providers who are working in the selected public health institutions.

### Study population

The randomly selected health care providers from the selected public health institution

### Inclusion criteria

All health care providers who are working in the selected public health institutions.

### Exclusion criteria

Those health care providers who are mentally/critically ill and on annual leave were excluded from the study.

### Sample size

The minimum sample size was determined by using a single population proportion formula [$n = [(Za/2)^2.P (1-P)]/d^2$] by assuming a 95% confidence level (Z a/2 = 1.96), a margin of error of 5%, P = proportion health care providers who are anxious in Southern Ethiopia (29.3%) [36] and a 5% addition for non-response rate. The final sample size became 334.

### Sampling technique and procedure

Six public hospitals were included in the study. The sample size in each hospital was allocated proportionally to the number of health professionals. The study participants were selected

using simple random sampling techniques. Within each hospital, the sample was taken from each department based on the proportion of their health professionals (Fig 1).

## Variables

**Dependent variable.**

- Anxiety, Depression, Stress.

**Independent variable.**

- Age, gender, religion, ethnicity, levels of education, marital status, job category, residence, monthly income, work experience, working setup, presence of infected colleague, presence of infected family members.

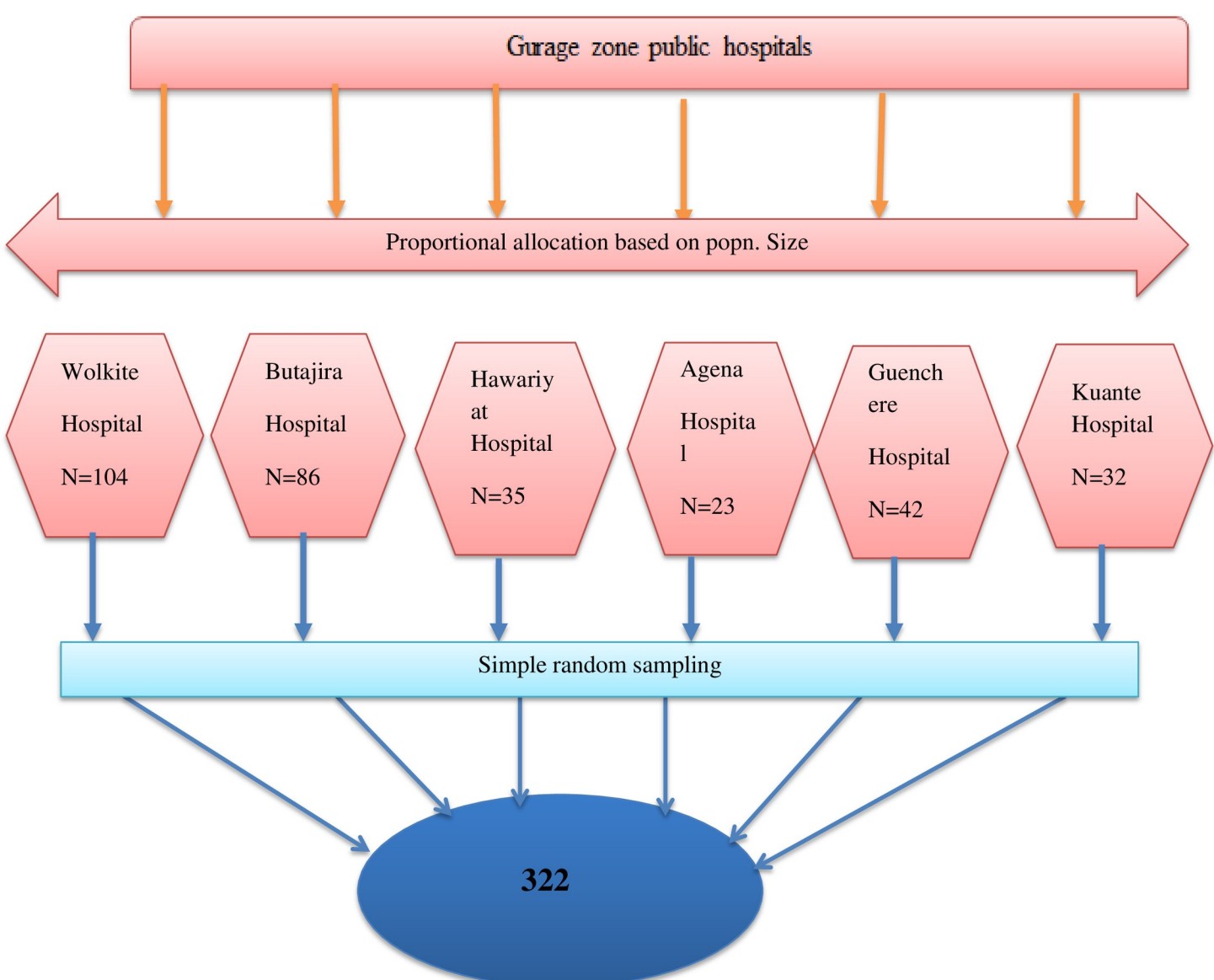

**Fig 1. The schematic presentation of the sampling procedure to select the study participants in Gurage zonal public hospital, SNNPR, Ethiopia, 2020 (n = 322).**

## Data collection instrument (tools) and procedure

Data were collected through a pre-tested, structured, and self-administered questionnaire to assess for symptoms of anxiety, depression, and stress using Amharic versions of validated and reliable measurement tools [37, 38].

The questionnaire consisted of five main themes: 1) demographics, which surveyed participants' socio-demographic information, including gender, age, educational status, marital status, ethnicity, and monthly income, 2) Occupational and personal-related characteristics of the participant such as job description, working setup, working experience, types of hospital, living condition, presence of suspected or confirmed colleagues and family members.

The third part comprised 7 items to assess the symptoms of anxiety by the Generalized Anxiety Disorder Scale (GAD 7-Scale) that contains a self-rated 7-item that asks for how often the participants have been bothered with the indicators over the past 2 weeks on a 4-point Likert scale, ranging from 0 (not at all) to 3 (nearly every day). The total GAD-7 score for the 7 items ranges from 0 to 21 and is classified as normal (0–4), mild (5–9), moderate (10–14), and severe (15–21) [39].

The fourth part comprised 9 items to assess symptoms of depression by the Patient Health Questionnaire (PHQ-9), a self-rated 9-item scale that asks if the participants have experienced symptoms of depression in the previous two weeks rated on a 4-point Likert-type scale ranging from 0 (not at all) to 3 (nearly every day). The total PHQ-9 scores range from 0 (absence of depressive symptoms) to 27 (most severe depressive symptoms) and classified into 0–4 = "Minimal depression," 5–9 = "Mild depression," 10–14 = "Moderate depression," 15–19 = "Moderately severe depression," and 20–27 = "Severe depression" [40, 41].

The fifth parts focus on the 10-item Perceived Stress Scale (PSS) which assesses the participants' perceived psychological stress by rating their feelings and thoughts during the past month. Participants are asked to rate their levels of agreement on a 5-point Likert-type scale ranging from 0 (never) to 4 (very often). It consists of two subscales, including a 6-item positive factor asking the ability to manage the stressors and a 4-item negative factor. The summation scores range from 0–40 with a higher score indicating a higher level of stress. The scores from 0–13 indicate low stress, whereas scores from 14–26 and 27–40 indicate moderate and high levels of stress, respectively [42]. The cutoff score for detecting clinically significant anxiety, depression, and stress were 7, 10, and 21, respectively [40, 43, 44]. Participants who had scores greater than the cutoff threshold were characterized as having severe symptoms.

## Data quality assurance and control

Data was collected from different healthcare workers in their respective wards using paper-based questionnaires. A questionnaire was developed and tested for reliability and validity and accordingly; the Cronbach alpha coefficient was found to be 0.88, 0.92, and 0.83 for anxiety, depression, and stress respectively. In addition, a pretest was done before actual data collection on 5% of a similar population in one hospital not included in actual data collection to assess flow, readability, and clarity of the questionnaire.

Eight data collectors and two supervisors were recruited for data collection, who have experience in data collection. To keep data quality supervisors and data collectors were oriented on how and what information they should collect from the targeted data sources. The completeness and consistency of the collected data were checked daily during data collection by the supervisor and the principal investigator. Whenever there appear incompleteness and ambiguity of recording, the filled information formats were crosschecked with source data soon. Individual records with incomplete data were also excluded.

## Data processing and analysis

The data was cleaned, coded, and entered into EpiData 3.1 and then exported to SPSS version 25.0 statistical package for further analysis. Data cleaning was performed to check for accuracy, consistencies, and missing values and variables.

Descriptive statistics and inferential statistics (chi-square tests) were carried out to illustrate the percentage and frequencies of study variables. Both bivariable and multivariable analyses were used to see the association of different variables. Those variables which revealed a statistically significant value at a p-value of ≤0.25 in the bivariable analysis were selected for multivariable logistic regression. For model fit, Hosmer and Lemeshow test was carried out and found to be 0.28, 0.398, and 0.587 for anxiety, depression, and stress respectively which indicated the final model was well fitted and multi-collinearity was also assessed. An adjusted odds ratio with a 95% confidence interval was used to measure the degree of association between variables. A P-value of ≤ 0.05 was considered statistically significant during multivariable logistic regression.

## Ethical considerations

Ethical Clearance approval was obtained from Wolkite University, Ethical Review Committee. Then data was collected after getting an official letter from the Zonal health department and permission from the medical director of each Hospital. The purpose of the study was explained to the study participants; anonymity, privacy, and confidentiality were ensured. Before data collection, informed verbal consent was obtained from the study participants. The respondents' right to refuse or withdraw from participating in the study was also fully acknowledged.
**Results**

## Socio-demographic characteristics of the respondents

There were 322 study participants involved in the study with a response rate of 96.5%. The highest proportion of respondents 157 (48.8%) were within the age group of 26–30 years with a mean age of 28.71 with SD 5.288. It showed that there was nearly equal participation of males (51.9%) and females (48.1). around two-thirds of the participants were Gurage (64.6%) followed by Amhara (17.1%) and Oromo (10.2%). Half of the participants were orthodox Christian and 56.5% of the participants were married. Regarding the educational status of the respondents, 57.8% (186) were degree holders followed by 34.2% (110) diploma (Table 1).

## Occupational and personal-related characteristics of the respondents

Concerning job description, more than one–thirds (35.1%) of the participants were nurses followed by the pharmacy (11.8%) and general practitioner (11.5%). Around two-thirds (66.1%) of the participants had ≤5 years of working experience with a mean experience of 4.7925 SD 3.58.one-thirds of the participants (34.5%) were employed at General Hospital. Nearly half of the participants (47.2%) of health care providers were living with their spouses. Most of the health care providers were practiced at the medical ward (13.3%) and POD followed by the emergency ward (11.8%) (Table 2).

## Prevalence of anxiety among health care providers in Gurage Zone, SNNPR, South West, Ethiopia, 2020

Table 3 shows the respondent's responses to the 7 items of the GAD-7. Through the past 2 weeks before the study. These respondents responded honestly to the following as occurring for several days, more than half the days, or nearly every day. worrying too much about

**Table 1. Socio-demographic information about health care providers in Gurage Zone, SNNPR, South West, Ethiopia, 2020.**

| Variables | Categories | Number | Percent |
|---|---|---|---|
| Age | 18–25 | 92 | 28.6 |
| | 26–30 | 157 | 48.8 |
| | 31–40 | 60 | 18.6 |
| | >40 | 13 | 4.0 |
| Sex | Male | 167 | 51.9 |
| | Female | 155 | 48.1 |
| Religion | Orthodox | 160 | 49.7 |
| | Muslim | 93 | 28.9 |
| | Protestant | 51 | 15.8 |
| | Catholic | 15 | 4.7 |
| | Other | 3 | .9 |
| Marital status | Married | 182 | 56.5 |
| | Single | 140 | 43.5 |
| Ethnicity | Gurage | 208 | 64.6 |
| | Oromo | 33 | 10.2 |
| | Amhara | 55 | 17.1 |
| | Tigre | 4 | 1.2 |
| | Others | 22 | 6.8 |
| Educational status | Diploma | 110 | 34.2 |
| | Degree | 186 | 57.8 |
| | Master's Degree and above | 26 | 8.1 |
| Average monthly income | Low | 142 | 44.1 |
| | High | 180 | 55.9 |

different things (52.5%); trouble relaxing (50.6%); feeling afraid as if something awful might happen (50.3%); being so restless that it is hard to sit still (48.8%); becoming easily annoyed or irritable (42.2%); not being able to stop or control worrying (38.8%); and feeling nervous, anxious or on edge (36.3%). Based on our findings 174(54%) of health care providers had minimal anxiety, 67(20.8%) had mild anxiety, 49(15.2%) had moderate anxiety and 32(9.9%) had severe anxiety.

For descriptive purposes only, a cutoff of ≥7 was used to distinguish severity for anxiety. so.116 (36%) of health care providers had a generalized anxiety disorder (Fig 2).

## Factors associated with anxiety among health care providers in Gurage Zone, SNNPR, South West, Ethiopia, 2020

Bivariable and multivariable logistic regression analysis was conducted to see the presence of association and to measure the relative effect of each independent variable on Generalized Anxiety Disorder among health care providers. Age, gender, religion, marital status, educational status, ethnicity, occupation, types of health facility, monthly income, experience, presence of infected colleague and family were significant factors associated with Anxiety among health care providers.

Among fitted variables included in the binary regression model for bivariable analysis, Age, religion, educational status, marital status, monthly income, experience, and presence of infected family were variables taken into consideration for multivariable analysis with $p$-value < 0.25. Under multivariable analysis, age, Educational status, monthly income, and presence of infected family were found to be statistically significant predictors of Anxiety among health care providers.

**Table 2. Occupational and personal-related characteristics of health care providers in Gurage Zone, SNNPR, South West, Ethiopia, 2020.**

| Variables | Categories | Number | Percent |
|---|---|---|---|
| Job description | Nurse | 113 | 35.1 |
| | Physicians | 44 | 13.7 |
| | Midwifery | 34 | 10.6 |
| | Pharmacy | 38 | 11.8 |
| | Lab Tech | 16 | 5.0 |
| | HO | 34 | 10.6 |
| | Environmental Health | 10 | 3.1 |
| | Others | 33 | 10.2 |
| Living condition | Spouse | 152 | 47.2 |
| | Family | 37 | 11.5 |
| | Friends | 7 | 2.2 |
| | Alone | 126 | 38.8 |
| Year of service (Experience) | ≤5 | 213 | 66.1 |
| | >5 | 109 | 33.9 |
| Working setup | Emergency | 38 | 11.8 |
| | Medical Ward | 43 | 13.3 |
| | Ophthalmology | 10 | 3.1 |
| | Surgical Ward | 25 | 7.7 |
| | Oby/Gyne ward | 33 | 10.2 |
| | Pediatric | 23 | 7.1 |
| | Medical Adult OPD | 43 | 13.3 |
| | Psychiatry OPD | 2 | .6 |
| | Dental Clinic | 1 | .3 |
| | Triage | 13 | 4.0 |
| | Pharmacy | 31 | 9.6 |
| | Laboratory | 15 | 4.6 |
| | OR | 8 | 2.5 |
| | ICU | 4 | 1.2 |
| | Others | 33 | 10.2 |
| Types of hospital | Primary | 85 | 26.4 |
| | General | 111 | 34.5 |
| | Referral | 47 | 14.6 |
| | Isolation center | 79 | 24.5 |
| Presence of infected colleagues | Yes | 96 | 29.7 |
| | No | 226 | 70.0 |
| Presence of infected family member | Yes | 38 | 11.8 |
| | No | 284 | 87.9 |

Health care providers whose age >40 years old were significantly more likely to develop anxiety than health care providers whose age 18–25 years old [AOR = 7.983; 95% CI (1.443–44.174)].

Based on educational status, respondents whose educational status masters and above were significantly more likely to develop anxiety than respondents whose educational status diploma [AOR = 3.243; 95% CI (1.003–10.482)].

Regarding monthly income, the odds of having anxiety were 1.87 times among respondents who had low monthly income as compared with those respondents who had a high monthly income [AOR = 1.868; 95% CI (1.140–3.061)]. Moreover, Health care providers who had

**Table 3. Prevalence of anxiety among health care providers in Gurage Zone, SNNPR, South West, Ethiopia, 2020.**

| Variables | Categories | | | |
|---|---|---|---|---|
| | Not at all number (%) | under half the days | over half the days | nearly every day |
| | | Number (%) | Number (%) | Number (%) |
| Feeling nervous, anxious, or on edge | 205 (63.7) | 61 (18.9) | 29 (9.0) | 27 (8.4) |
| Not being able to stop or control worrying | 197 (61.2) | 63 (19.6) | 31 (9.6) | 31 (9.6) |
| Worrying too much about different things | 153 (47.5) | 84 (26.1) | 50 (15.5) | 35 (10.9) |
| Trouble relaxing | 159 (49.4) | 70 (21.7) | 59 (18.3) | 34 (10.6) |
| Being so restless that it's hard to sit still | 165 (51.2) | 83 (25.8) | 44 (13.7) | 30 (9.3) |
| Becoming easily annoyed or irritable | 186 (57.8) | 75 (23.3) | 37 (11.5) | 24 (7.5) |
| Feeling afraid as if something awful might happen | 160 (49.7) | 75(23.3) | 55(17.1) | 32(9.9%) |

infected family members were significantly more likely to develop anxiety than respondents who didn't have infected family members [AOR = 3.296; 95% CI (1.503–7.227)] (Table 4).

## Prevalence of depression among health care providers in Gurage Zone, SNNPR, South West, Ethiopia, 2020

Table 5 shows the respondent's responses to the 9 items of the PHQ-9. Through the past 2 weeks before the study. these providers replied honestly to the following as occurring for

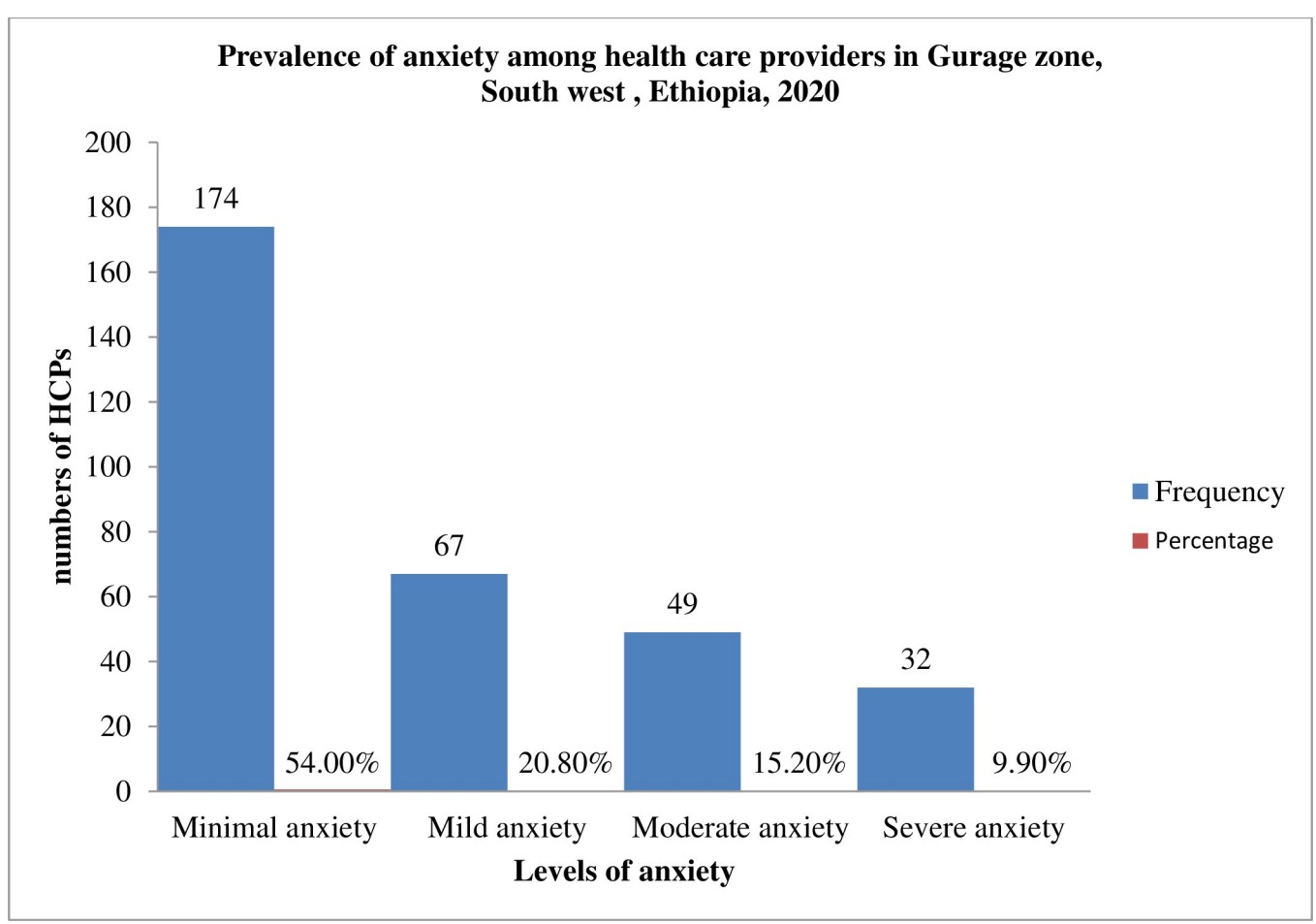

**Fig 2. Prevalence of anxiety among health care providers in Gurage Zone, SNNPR, South West, Ethiopia, 2020 (n = 322).**

**Table 4. Factors associated with anxiety among health care providers in Gurage Zone health institutions, SNNPR, South West, Ethiopia, 2020.**

| Variable | Levels of Anxiety | | COR(95%, CI) | AOR(95%, CI) |
|---|---|---|---|---|
| | No Anxiety n (%) | Anxiety n (%) | | |
| Age in year | | | | |
| 18–25 | 65(20.18) | 27(8.38) | 1.00 | 1.00 |
| 26–30 | 111(34.47) | 46(12.8) | 0.998(0.567–1.756) | 1.064(0.587–1.929) |
| 31–40 | 28(8.43) | 32(9.93) | 2.751(1.398–5.416)* | 2.019(0.940–4.339) |
| >40 | 2(0.62) | 11(3.4) | 13.241(2.749–63.774)* | 7.983(1.443–44.174)** |
| Monthly income | | | | |
| Low | 80(24.8) | 62(19.2) | 1.808(1.142–2.865)* | 1.868(1.140–3.061)** |
| High | 126(39.1) | 54(16.7) | 1.00 | 1.00 |
| Educational status | | | | |
| Diploma | 73(22.67) | 37(11.49) | 1.00 | 1.00 |
| Degree | 126(39.13) | 60(18.63) | 0.940(0.569–1.550) | 0.998(0.570–1.747) |
| Masters and above | 7(2.1) | 19(5.9) | 5.355(2.066–13.883)* | 3.243(1.003–10.482)** |
| Infected family member | | | | |
| No | 194(60.25) | 90(27.95) | 1.00 | 1.00 |
| Yes | 12(3.72) | 26(8.06) | 4.670(2.255–9.674)* | 3.296(1.503–7.227)** |

*p value<0.05 at bivariate logistic regression

**p value<0.05 at multivariate logistic regression.

several days, more than half the days, or nearly every day: feeling tired or having little energy (42.5%); poor appetite or overeating (39.4%); little interest or pleasure in doing things (39.1%); moving or speaking so slowly that other people could have noticed; so fidgety or restless that you have been moving around a lot more than usual (37.6%); trouble falling or staying asleep or sleeping too much (37.3%); feeling bad about yourself or that you are a failure or have let yourself or your family down (34.2%); feeling down, depressed, or hopeless (31.4%); thoughts that you would be better off dead or of hurting yourself in some way (30.7%) and trouble concentrating on things, such as reading the newspaper or watching television (30.1%). Based on our finding three-fifth (60.2%) of health care providers had minimal depression, 45(14%) had

**Table 5. Prevalence of depression among health care providers in Gurage Zone, SNNPR, South West, Ethiopia, 2020.**

| Variables | Categories | | | |
|---|---|---|---|---|
| | Not at all number (%) | under half the days | over half the days | nearly every day |
| | | Number (%) | Number (%) | Number (%) |
| Little interest or pleasure in doing things | 196 (60.9) | 50 (15.5) | 42 (13.0) | 34 (10.6) |
| Feeling down, depressed, or hopeless | 221 (68.6) | 47(14.6) | 38 (11.8) | 16 (5.0) |
| Trouble falling or staying asleep, or sleeping too much | 202 (62.7) | 55 (17.1) | 43(13.4) | 22 (6.8) |
| Feeling tired or having little energy | 185 (57.5) | 69 (21.4) | 45 (14.0) | 23 (7.1) |
| Poor appetite or over eating | 195 (60.6) | 66 (20.5) | 41 (12.7) | 20 (6.2) |
| Feeling bad about yourself or that you are a failure or have let yourself or your family down | 212 (65.8) | 73 (22.7) | 24 (7.5) | 13 (4.0) |
| Trouble concentrating on things, such as reading the newspaper or watching television | 196 (69.9) | 53(16.5) | 50(15.5) | 23(7.1%) |
| Moving or speaking so slowly that other people could have noticed? Or the opposite—being so fidgety or restless that you have been moving around a lot more than usual | 201 (62.4%) | 66 (20.5%) | 36 (11.2%) | 19 (5.9%) |
| Thoughts that you would be better off dead or of hurting yourself in some way | 223 (69.3%) | 58 (18.0%) | 32 (9.9%) | 9 (2.8%) |

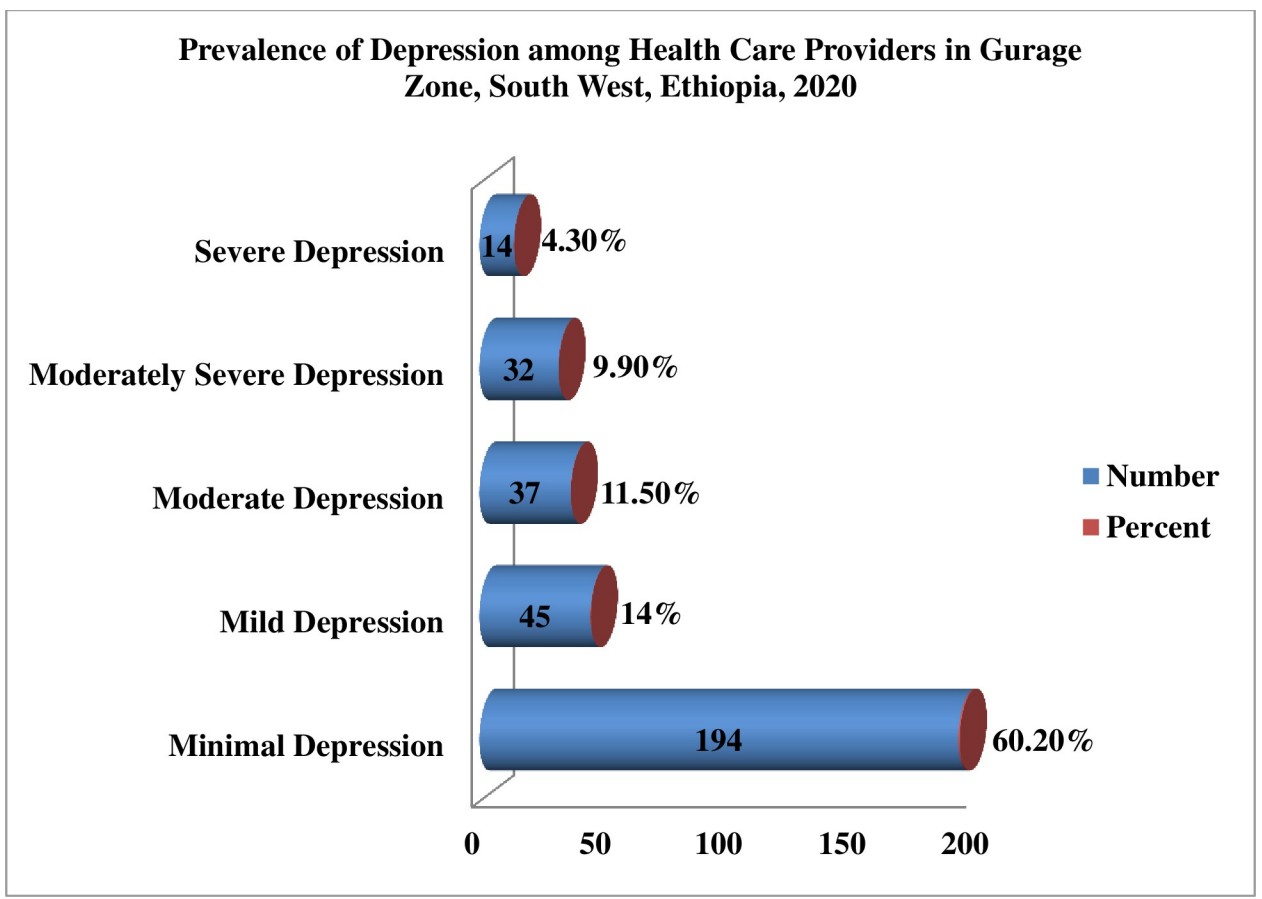

**Fig 3. Prevalence of depression among health care providers in Gurage Zone, SNNPR, South West, Ethiopia, 2020 (n = 322).**

mild depression, 37(11.5%) had moderate anxiety, 32(9.9%) had moderately severe depression and 14(4.3%) had severe depression.

For descriptive purposes only, a cutoff of ≥10 was used to distinguish the severity of depression. So. One-fourths of (25.8%) of health care providers had depression related to covid-19 pandemics (Fig 3).

## Factors associated with depression among health care providers in Gurage Zone, SNNPR, South West, Ethiopia, 2020

First Bivariable logistic regression analysis was conducted to detect the presence of association and measure the relative effect of each independent variable against depression. As a result, among all other variables. Age, gender, marital status, educational status, ethnicity, occupation, types of health facility, living condition, monthly income, experience, and presence of infected colleague were found to have an association (i.e. $p$-value of $< 0.25$) and become eligible for multivariable analysis. Then, the multivariable logistic regression analysis showed that gender, educational status, living condition, occupation, and presence of infected family were found to be statistically significant predictors of depression among health care providers.

The odds of having depression were 1.9 times among male health care providers as compared with female health care providers [AOR = 1.908; 95% CI (1.040–3.500)].

Based on educational status, the odds of having depression among providers whose educational status masters and above were 10.8 times and degrees were 2.26 times as compared with

**Table 6. Factors associated with depression among health care providers in Gurage Zone health institutions, SNNPR, South West, Ethiopia, 2020.**

| Variable | Levels of Depression | | COR(95%, CI) | AOR(95%, CI) |
|---|---|---|---|---|
| | No Depression | Depression | | |
| | n (%) | n (%) | | |
| **Gender** | | | | |
| Male | 117(36,33) | 50(15.52) | 1.580(0.951–2.624) | 1.908(1.040–3.500)** |
| Female | 122(37.88) | 33(10.24) | 1.00 | 1.00 |
| **Educational status** | | | | |
| Diploma | 90(27.95) | 20(6.2) | 1.00 | 1.00 |
| Degree | 138(42.85) | 48(14.90) | 1.565(0.872–2.811) | 2.269(1.131–4.551)** |
| Masters and above | 11(3.41) | 15(4.65) | 6.136(2.454–15.345)* | 10.844(3.314–35.482)** |
| **Living status** | | | | |
| Husband | 104(32.29) | 48(14.90) | 2.308(1.292–4.122)* | 5.824(1.896–17.888)** |
| Family | 24(7.45) | 13(4.03) | 2.708(1.191–6.159)* | 3.938(1.380–11.242)** |
| Friend | 6(1.86) | 1(0.31) | 0.833(0.095–7.286) | 0.641(0.063–6.538) |
| Alone | 105(32.06) | 21(6.52) | 1.00 | 1.00 |
| **Occupation** | | | | |
| Nurse | 85(26.39) | 28(8.7) | 1.00 | 1.00 |
| Physician | 38(11.8) | 6(1.86) | 0.479(0.183–1.253) | 0.197(0.60–0.651)** |
| Midwifery | 28(8.7) | 6(1.86) | 0.651(0.244–1.733) | 0.846(0.291–2.458) |
| Pharmacy | 19(5.9) | 19(5.9) | 3.036(1.411–6.530)* | 4.519(1.880–11.006)** |
| Lab technician | 13(4.03) | 3(0.93) | 0.701(0.186–2.638) | 1.303(0.307–5.522) |
| Health officer | 27(8.38) | 7(2.17) | 0.787(0.309–2.004) | 0.543(0.180–1.642) |
| Environmental Health | 6(1.86) | 4(1.24) | 2.024(0.532–7.693) | 0.716(0.142–3.593) |
| Others | 23(7.14) | 10(3/1) | 1.320(0.560–3.108) | 0.787(0.281–2.202) |

*p value<0.05 at bivariate logistic regression

**p value<0.05 at multivariate logistic regression.

those respondents whose educational status were diploma [AOR = 10.844; 95% CI (1.131–4.551)], and [AOR = 2.269; 95% CI (3.314–35.482)] respectively.

Health care providers who live with their husband/wife and those respondents who live with their families were significantly more likely to develop depression than health care providers who live alone [AOR = 5.824; 95% CI (1.896–17.888)] and [AOR = 3.938; 95% CI (1.380–11.242)] respectively. On the other hand, the odds of having depression among pharmacists were 4.5 times and among physicians were 0.2 times as compared with nurses, [AOR = (4.519; 95% CI (1.880–11.006)] and [AOR = (0.197; 95% CI (0.60–0.651))] respectively (Table 6).

## Prevalence of perceived stress among health care providers in Gurage Zone, SNNPR, South West, Ethiopia, 2020

Table 7 shows the respondents' responses to the 10 Item perceived stress scale (PSS) during the last month before the study. These providers replied honestly to the following as occurring for sometimes, fairly often, or very often. Felt difficulties were piling up so high that you could not overcome them (64.0%); been upset because of something that happened unexpectedly (58.4%); been angered because of things that were outside of your control (54.3%); found that you could not cope with all the things that you had to do (46.9%); felt that you were unable to control the important things in your life (45.9%); felt nervous and "stressed" (41.3%); been

**Table 7. Prevalence of perceived stress among health care providers in Gurage Zone, SNNPR, South West, Ethiopia, 2020.**

| Variables | Categories | | | | |
|---|---|---|---|---|---|
| | Never (%) | Almost never (%) | Sometimes (%) | Fairly Often (%) | Very Often (%) |
| Been upset because of something that happened unexpectedly? | 89 (27.6) | 45 (14.0) | 52 (16.1) | 62 (19.3) | 74 (23.0%) |
| Felt that you were unable to control the important things in your life? | 112 (34.8) | 62 (19.3) | 71 (22.0) | 45 (14.0) | 32 (9.9) |
| Felt nervous and "stressed"? | 119 (37.0) | 70 (21.7) | 64(19.9) | 46 (14.3) | 23 (7.1) |
| **Felt confident about your ability to handle your problems?** | 53 (16.5) | 74 (23.0) | 70 (21.7) | 68 (21.1) | 57 (17.7) |
| **Felt that things were going your way?** | 41 (12.7) | 61 (18.9) | 79 (24.5) | 86(26.7) | 55 (17.1) |
| Found that you could not cope with all the things that you had to do? | 51 (15.8) | 120 (37.3) | 75 (23.3) | 59 (18.3) | 17 (5.3) |
| **Been able to control irritations in your life?** | 49 (15.2) | 147 (45.7) | 64 (19.9) | 34 (10.6%) | 28 (8.7) |
| **Felt that you were on top of things?** | 47 (14.6) | 131(40.7%) | 54 (16.8%) | 53 (16.5%) | 37 (11.5) |
| Been angered because of things that were outside of your control? | 53 (16.5%) | 94 (29.2%) | 65 (20.2%) | 60 (18.6%) | 50 (15.5%) |
| Felt difficulties were piling up so high that you could not overcome them? | 65 (20.2) | 51 (15.8) | 58 (18.0) | 69 (21.4) | 79 (24.5) |

The bold letter indicates a 4-item negative factor asking the ability to manage the stressors.

able to control irritations in your life (39.1%); felt that you were on top of things (44.7%); felt confident about your ability to handle your problems (60.5%) and felt that things were going your way (68.4%).

Based on our finding three-fifth (60.2%) of health care providers had low stress, 45(14%) had moderate stress, and 14(4.3%) had high levels of stress.

The summation scores range from 0–40 with a higher score indicating a higher level of stress. The scores from 0–13 indicate low stress, whereas scores from 14–26 and 27–40 indicate moderate and high levels of stress, respectively.

For descriptive purposes only, a cutoff of ≥21 was used to distinguish the severity of stress. So. 101 (31.4%) of HCPs had stress whereas 221 (68.6%) of HCPs had no stress related to covid-19 pandemics.

## Factors associated with stress among health care providers in Gurage Zone, SNNPR, South West, Ethiopia, 2020

Bivariate and multivariate logistic regression analysis was conducted to see the presence of association and to measure the relative effect of each independent variable on overall perceived stress among health care providers. Age, gender, marital status, educational status, ethnicity, occupation, types of health facility, monthly income, experience, presence of infected colleague and family were significant factors associated with stress among health care providers.

Among fitted variables included in the binary regression model for bivariate analysis, gender, marital status, ethnicity, occupation, types of health facility, monthly income and experience were variables taken into consideration for multivariate analysis with a *p*-value < 0.25. Under multivariate analysis type of health facility and monthly income were found to be statistically significant predictors of stress among health care providers.

Health care providers who are working at general and referral hospitals were significantly more likely to develop stress than health care providers who were working at primary hospitals [AOR = 4.835; 95% CI (2.189–10.680)], and [AOR = 3.263; 95% CI (1.302–8.178)] respectively.

Health care providers who had low monthly income were significantly more likely to develop stress than health care providers who had high monthly income [AOR = 2.289; 95% CI (1.349–3.885)] (Table 8).

**Table 8. Factors associated with stress among health care providers in Gurage Zone health institutions, SNNPR, South West, Ethiopia, 2020.**

| Variable | Levels of Stress | | COR(95%, CI) | AOR(95%, CI) |
|---|---|---|---|---|
| | No Stress n (%) | Stress n (%) | | |
| **Types of health facility** | | | | |
| Primary | 75(23.29) | 10(3.1) | 1.00 | 1.00 |
| General | 64(19.88) | 47(14.6) | 5.508(2.577–11.774)* | 4.835(2.189–10.680)** |
| Referral | 31(9.63) | 16(4.97) | 3.871(1.583–9.465)* | 3.263(1.302–8.178)** |
| Isolation center | 51(15.84) | 28(8.7) | 4.118(1.841–9.209)* | 5.270(2.275–12.209)** |
| **Monthly income** | | | | |
| Low | 85(26.3) | 57(17.7) | 2.073(1.286–3.342)* | 2.289(1.349–3.885)** |
| High | 136(42.2) | 44(13.67) | 1.00 | 1.00 |

*p value<0.05 at bivariate logistic regression

**p value<0.05 at multivariate logistic regression.

Generally, 36.0%, 25.80%, and 31.4% of health care providers in the Gurage zonal institution had anxiety, depression, and stress respectively (Fig 4).

## Discussion

The results of this study had shown that the overall prevalence of anxiety among health care providers in Gurage Zonal Public hospital was 36%, [95% CI = (30.7%- 41.3%)] which is in

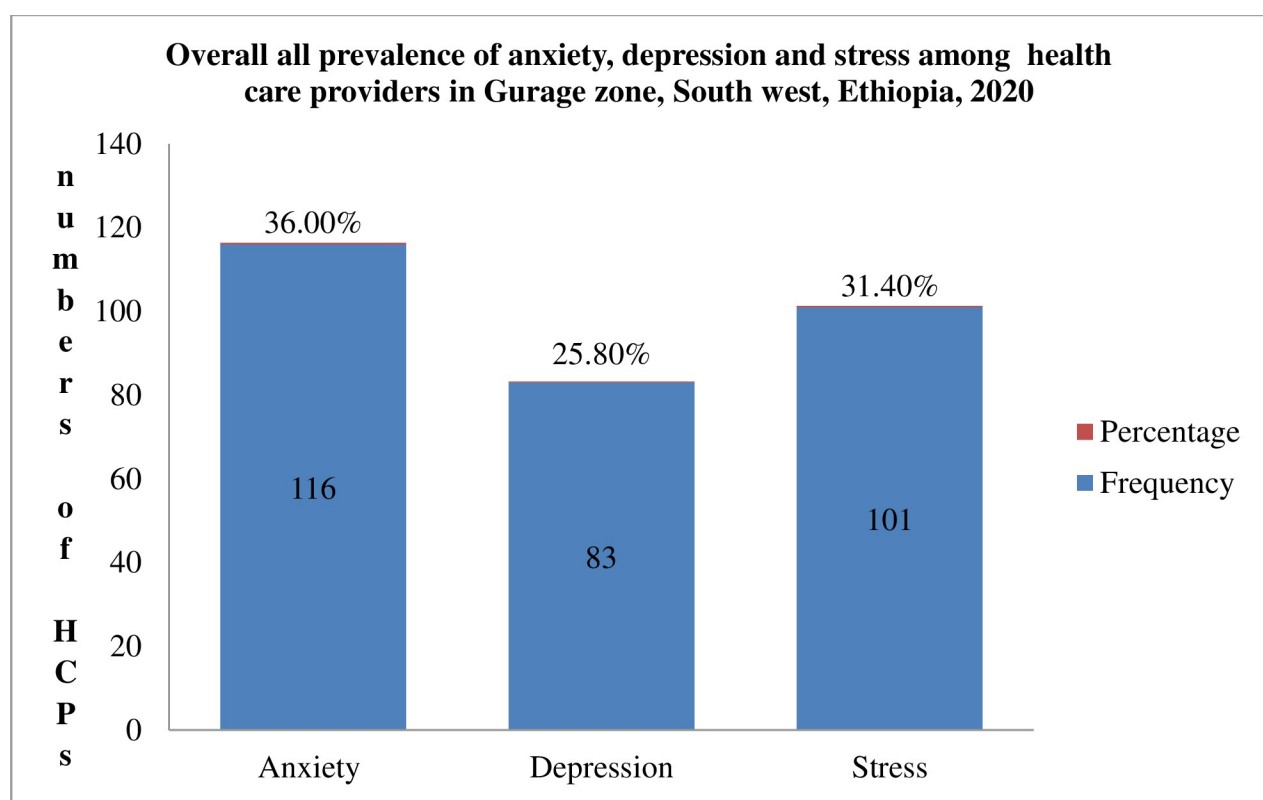

**Fig 4. Overall prevalence of anxiety, depression, and stress among HCPs in Gurage Zonal Health Institution's, SNNPR, Ethiopia, 2020 (n = 322).**

line with the previously reported study from Ethiopia(29.3%) [45], Nepal (38%) [46], India (31.6) [47].However, it is less than the findings in Gondar (69.6%) [48], Debre Tabor (63%) [49], Central Ethiopia (78%) [50], Addis Ababa (51.6) [51], Egypt, and Saudi Arabia (58.9%) [52]). Mali (73.3%) [53], Saudi Arabia (51.4%) [54], (78.8%) [55], Iran (67.55%) [56], Iraq (48.9%) [57], Nepal (41.9%) [58], Turkey (45.1%) [59], China (41.1%) [60], (44.6%) [4], North Italy (50.1%) [61], and USA (43%) [62]. However, it is higher than the study conducted in St Peter hospital, Addis Ababa (21.2%) [63], St Paul Hospital (21.9%) [64], multinational multi-center study (8.7%) [65], Middle East country (23.6%) [66], Iran (25.8%) [67], China 8.5% [68] (14.3%) [69], India (17.7%) [47], Italy (19.8%) [70], Malaysia (29.7%) [71], and Australia (21%). This discrepancy may be due to differences in workload, socioeconomic, cultural, and environmental factors, variation in the availability of personal protective equipment and resources, the difference in emotional response related to previous experience/exposure with SARS, MERS, and H1N1 epidemics, and also it may be related to the different cut-off scores used to define levels of clinically significant anxiety.

According to this study being older than 40 years is significantly associated with anxiety. The finding is also supported by a study reported from Debre tabor [49], Saudi Arabia [72, 73]. South Korea [74] and India [75]. This may be due to older health care providers are among the most affected by the COVID-19 pandemic in terms of illness severity and mortality, increased risk of transmission more prone to complications, and they could also live with young children and/or have older people in their extended family, which could cause them to worry about bringing the virus home to their family members as well as older health care workers tend to have lower stress reactivity, poor emotional regulation and well-being than younger HCPs. Moreover, they are also more likely to suffer psychological impacts such as anxiety due to isolation, heavy workload, and facts about the COVID-19 pandemic which is complicated by pre-existing physical health problems, medical comorbidities, and existing mental health symptoms.

Our finding also showed that having infected family members is significantly associated with anxiety. It is supported by a previous study conducted in Debre tabor [49], Gondar [76], and China [77]. This anxiety might arise from close family relationships and concerns about family members' health conditions, the absence of specific treatments for COVID-19 during the initial periods of the pandemic, and isolation from their loved ones during quarantine for prolonged periods.

Our study also revealed that those health care providers who had low monthly income were significantly associated with anxiety. This is supported by research done in St Peter hospital [63]. This may due to preoccupation with fear of how to cope with the potential economical challenge faced during the pandemic and increased psychological and economic pressure resulting from socioeconomic challenges that may critically impact mental health.

Our study finding showed that 25.8% [95% CI = (21.1%- 30.4%)] of Gurage Zone health care providers are suffering from depression during COVID-19 outbreak which is in line with study done in Iran(24.3%) [67], Middle East counties (27.4%) [66], Saudi Arabia (26.1%) [55], India 25%) [78], Turkey (23.6%) [59], Australia (27.6%), Italy (24.73%, 26%) [61, 70], and USA (26%) [62].This finding is higher than the study conducted in St Paul Hospital, Ethiopia (20.2%) [79], multinational multi-center study (5.3%) [65], India (11.4%) [47], China (9.5%) [68], (10.7%) [69]. However, it is lower than in study carried out in Gondar (55.3%) [48], St Peter Hospital, Addis Ababa (36.5%) [63], Central Ethiopia (60.3%) [50], Mali (71.9%) [53], Egypt and Saudi Arabia (69%) [52], (55.2%) [54], systematic review in Iran (55.89%) [80], Oman (32.3%) [81], Nepal (37.5%) [58], Malaysia (31%) [71], China (46.9%) [60], (50%) [4]. The discrepancy could be explained by the difference in socioeconomic status, social support, study setting, variability of health care workers, sample size variation, the difference in

methods, environmental and organization culture, as well as social and cultural issues, which might contribute to this difference.

Our study revealed that Males health care providers were about two times more likely to become depressed than females. This is in line with the study carried out in India [78]. This might be due to our socio-cultural norms males HCPs had an economic burden which is expected to help other members of the family and relatives as a result, they are more prone to financial deprivation, which leads to developing depression than their counterparts. But this finding is inconsistent with other studies conducted in Egypt and Saudi Arabia [52], Low and Middle-income countries [82], Middle East countries [66], Turkey [59], Iran [83], India [47, 84], Poland [85], Italy [70, 86], and United Kingdom [87] which states female HCPs were more prone to depression due to females being more commonly exposed to mental illness, cultural factors, and hormonal fluctuations.

Our finding also revealed that participants with high educational levels were more depressed than those with lower educational statuses. This could be related to the increments of workload to those who had higher educational attainment and they conducted and explored different types of scientific researches about the virulent nature of the COVID-19 pandemic which induces depression.

According to our study, Health care providers living with their spouse and family were more likely to develop depression than those HCPs living alone. This finding is supported by research findings in St Paul, Ethiopia [64], Saudi Arabia [55], India [75, 84], and the United Kingdom [87]. which states that HCWs who were either married or married with children were more depressed than those among unmarried HCWs/ living alone [84]. The possible explanation could be primary worry of all HCPs was the safety of their families during the COVID-19 pandemic, which was regarded as a major depressive factor. Furthermore, married HCPs were found to be more hopeless, concern for family members and their wellbeing could contribute to their feeling of hopelessness.

The results of our study found that Nurse and Pharmacists were more likely to develop depression. Similar results were reported in research conducted in Mali [53], Middle East countries [66], China [69, 88], and Brazil [89]. The possible explanation might be due to nurses are frontline healthcare workers, directly engaged in diagnosis, treatment, and care of patients with COVID-19 and they have long work shifts and closer contact with patients, by undertaking most of the tasks related to infectious disease containment, as a result, they are prone to fatigue, tension, and depression. Besides these in a study conducted in low and middle-income countries (LMICs) showed that Nurses and other HCPs in non-physician roles experienced greater depressive symptom severity compared to HCPs in physician roles [82]. The pharmacist is also involved directly to provide drugs for patients with COVID-19 and they are at high risk for developing depression.

Our study finding showed that during the COVID-19 pandemic, 31.4% [95% CI = (26.4%-36.0%)] of HCPs had stress. Which is congruent with study finding in Central Ethiopia (33.8%) [50], benchi-sheko (32.5%) [90], The finding is less than study reported from Addis Ababa (43.4%) [91]), Dilla, Ethiopia (51.6%) [43], Egypt and Saudi Arabia (55.9%) [52], a systematic review in Iran (45%) [67], (62.9%) [80], China (69.1%) [60], and USA (80.1%) [62]. However, it is higher than in study done in Gondar (20.5%) [48], multinational multi-center study (2.2%) [65], Oman (23.8%) [81], Malaysia (23.5%) [71], and Italy(21.90%, 8.9%) [70, 92].

Health care providers who are working at general and referral hospitals were more likely to become stressed than health care providers who were working at primary hospitals. This is supported by a systematic review and meta-analysis conducted in china which states that a considerable proportion of healthcare workers within secondary and tertiary hospitals developed adverse psychological outcomes during the COVID-19 pandemic [32]. A similar study

conducted in China showed that those HCPs working in secondary hospitals reported high levels of psychological problems [4].

In this study HCPs who had low monthly income were more likely to be stressed than those HCPs who had high monthly income. This is supported by research conducted in Addis Ababa, Ethiopia. This could be due to the socioeconomic impact of a virus might be much significant to the extent of unable to buy safety measures of prevention, such as facemask, soaps, and sanitizers. In addition during this pandemic period, they were not able to fulfill their basic needs of day-to-day life [93].

## Strength of the study

We have used a previously validated and well-established instrument to measure our outcome variables GAD-7, PHQ, and PSS for the assessment of anxiety, depression, and stress respectively. Moreover, there was a proportionate representation of health care providers from each department in this study; this would likely mitigate the bias of having a higher number of nurses/ doctors as in previous studies conducted in other regions of Ethiopia.

## Limitation of the study

The study has certain limitations which must be acknowledged. First, we did not explore the common risk factors for anxiety, depression, and stress, like a history of anxiety, depression, and stress, comorbidities like chronic diseases, social support, and communication. Second, responses to the survey were self-reported. It may have resulted in reporting biases for social desirability which may have affected the results and finally, this study cannot show cause and effect relationship since it is a cross-sectional type. Despite the identified limitations, these results contribute to the information relating to the overwhelming problem faced by HCPs especially related to the commonly encountered mental health problems while caring not only in Ethiopia but also at the global level.

## Conclusion and recommendation

Based on our findings, significant numbers of healthcare workers were suffered from anxiety, depression, and stress during the COVID-19 outbreak. On most occasions, the mental health impact of a disease outbreak is usually neglected during pandemic management although the consequences are costly. Therefore, the Federal Ministry of Health in collaboration with hospitals should pass emergency legislation to protect healthcare providers who are at high risk of exposure to mental health problems. This should include financial protections for healthcare providers who contract COVID-19 and supplement additional safety requirements for healthcare facilities. Moreover, mental health professionals should pay attention to the role of family members' health and monitor the mental wellbeing of health care providers too.

The finding revealed that age, educational status, monthly income, and the presence of infected families were statistically associated with anxiety. Besides this, gender, educational status, living condition, and occupation were found to be statistically significant predictors of depression among health care providers. Our study finding also showed that type of health facility and monthly income were found to be statistically significant predictors of stress among health care providers.

## Supporting information

**S1 File.**
(SAV)

## Acknowledgments

We would like to express our heartfelt gratitude to the Gurage zonal health department, hospital administration staff, and health care providers for their valuable supports while collecting the data. And our thanks go to go to all data collectors and supervisors for their endeavor.

## Author Contributions

**Conceptualization:** Fisha Alebel GebreEyesus, Tadesse Tsehay Tarekegn, Baye Tsegaye Amlak, Bisrat Zeleke Shiferaw, Mamo Solomon Emeria, Omega Tolessa Geleta, Tamene Fetene Terefe, Mtiku Mammo Tadereregew, Melkamu Senbeta Jimma, Fatuma Seid Degu, Elias Nigusu Abdisa, Menen Amare Eshetu, Natnael Moges Misganaw, Ermias Sisay Chanie.

**Data curation:** Fisha Alebel GebreEyesus.

**Formal analysis:** Fisha Alebel GebreEyesus, Tadesse Tsehay Tarekegn, Baye Tsegaye Amlak, Mamo Solomon Emeria, Tamene Fetene Terefe, Ermias Sisay Chanie.

**Funding acquisition:** Fisha Alebel GebreEyesus, Tadesse Tsehay Tarekegn, Bisrat Zeleke Shiferaw, Omega Tolessa Geleta, Tamene Fetene Terefe, Mtiku Mammo Tadereregew, Melkamu Senbeta Jimma, Fatuma Seid Degu, Elias Nigusu Abdisa, Menen Amare Eshetu, Natnael Moges Misganaw.

**Investigation:** Fisha Alebel GebreEyesus, Tadesse Tsehay Tarekegn, Baye Tsegaye Amlak, Bisrat Zeleke Shiferaw, Mamo Solomon Emeria, Omega Tolessa Geleta, Tamene Fetene Terefe, Mtiku Mammo Tadereregew, Melkamu Senbeta Jimma, Fatuma Seid Degu, Elias Nigusu Abdisa, Menen Amare Eshetu, Natnael Moges Misganaw, Ermias Sisay Chanie.

**Methodology:** Fisha Alebel GebreEyesus, Tadesse Tsehay Tarekegn, Baye Tsegaye Amlak, Bisrat Zeleke Shiferaw, Mamo Solomon Emeria, Omega Tolessa Geleta, Tamene Fetene Terefe, Mtiku Mammo Tadereregew, Melkamu Senbeta Jimma, Fatuma Seid Degu, Elias Nigusu Abdisa, Menen Amare Eshetu, Natnael Moges Misganaw, Ermias Sisay Chanie.

**Resources:** Fisha Alebel GebreEyesus, Ermias Sisay Chanie.

**Software:** Fisha Alebel GebreEyesus, Baye Tsegaye Amlak, Bisrat Zeleke Shiferaw, Mamo Solomon Emeria, Omega Tolessa Geleta, Tamene Fetene Terefe, Mtiku Mammo Tadereregew, Melkamu Senbeta Jimma, Fatuma Seid Degu, Elias Nigusu Abdisa, Menen Amare Eshetu, Natnael Moges Misganaw, Ermias Sisay Chanie.

**Supervision:** Fisha Alebel GebreEyesus, Tadesse Tsehay Tarekegn, Baye Tsegaye Amlak, Bisrat Zeleke Shiferaw, Mamo Solomon Emeria, Omega Tolessa Geleta, Tamene Fetene Terefe, Mtiku Mammo Tadereregew, Melkamu Senbeta Jimma, Fatuma Seid Degu, Elias Nigusu Abdisa, Menen Amare Eshetu, Natnael Moges Misganaw, Ermias Sisay Chanie.

**Validation:** Fisha Alebel GebreEyesus, Tadesse Tsehay Tarekegn, Baye Tsegaye Amlak, Bisrat Zeleke Shiferaw, Mamo Solomon Emeria, Omega Tolessa Geleta, Tamene Fetene Terefe, Mtiku Mammo Tadereregew, Melkamu Senbeta Jimma, Fatuma Seid Degu, Elias Nigusu Abdisa, Menen Amare Eshetu, Natnael Moges Misganaw, Ermias Sisay Chanie.

**Visualization:** Fisha Alebel GebreEyesus, Elias Nigusu Abdisa, Ermias Sisay Chanie.

**Writing – original draft:** Fisha Alebel GebreEyesus, Tadesse Tsehay Tarekegn, Baye Tsegaye Amlak, Bisrat Zeleke Shiferaw, Mamo Solomon Emeria, Omega Tolessa Geleta, Tamene Fetene Terefe, Mtiku Mammo Tadereregew, Melkamu Senbeta Jimma, Fatuma Seid Degu,

Elias Nigusu Abdisa, Menen Amare Eshetu, Natnael Moges Misganaw, Ermias Sisay Chanie.

**Writing – review & editing:** Fisha Alebel GebreEyesus, Tadesse Tsehay Tarekegn, Baye Tsegaye Amlak, Bisrat Zeleke Shiferaw, Mamo Solomon Emeria, Omega Tolessa Geleta, Tamene Fetene Terefe, Mtiku Mammo Tadereregew, Melkamu Senbeta Jimma, Fatuma Seid Degu, Elias Nigusu Abdisa, Menen Amare Eshetu, Natnael Moges Misganaw, Ermias Sisay Chanie.

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
