## [Decision Letter · Decision Letter 0]

12 Aug 2021

PONE-D-21-20584

Levels and predictors of anxiety, depression, and stress during COVID-19 pandemic among frontline healthcare providers in Gurage zonal public hospitals, Southwest Ethiopia, 2020: A Multicenter Cross-Sectional Study

PLOS ONE

Dear Dr. GebreEyesus,

Thank you for submitting your manuscript to PLOS ONE. After careful consideration, we feel that it has merit but does not fully meet PLOS ONE’s publication criteria as it currently stands. Therefore, we invite you to submit a revised version of the manuscript that addresses the points raised during the review process.

We look forward to receiving your revised manuscript.

Kind regards,

Kensaku Kasuga

Academic Editor

PLOS ONE

Journal Requirements:

2. Please include your tables as part of your main manuscript and remove the individual files. Please note that supplementary tables should be uploaded as separate "supporting information" files

4. We note you have included a table to which you do not refer in the text of your manuscript. Please ensure that you refer to Table 5 in your text; if accepted, production will need this reference to link the reader to the Table

5. We noticed you have some minor occurrence of overlapping text with the following previous publication(s), which needs to be addressed:

- https://www.dovepress.com/the-psychological-impact-of-covid-19-outbreak-on-nurses-working-in-the-peer-reviewed-fulltext-article-PRBM#:~:text=Due%20to%20this%20nurses%20with,associated%20with%20stress%20amon

The text that needs to be addressed involves the Discussion.

In your revision ensure you cite all your sources (including your own works), and quote or rephrase any duplicated text outside the methods section. Further consideration is dependent on these concerns being addressed.

Additional Editor Comments:

Please respond to the reviewer's comments.

Reviewers' comments:

Reviewer's Responses to Questions

**Comments to the Author**

1. Is the manuscript technically sound, and do the data support the conclusions?

Reviewer #1: Yes

2. Has the statistical analysis been performed appropriately and rigorously? 

Reviewer #1: Yes

3. Have the authors made all data underlying the findings in their manuscript fully available?

Reviewer #1: Yes

4. Is the manuscript presented in an intelligible fashion and written in standard English?

Reviewer #1: Yes

5. Review Comments to the Author

Reviewer #1: GebreEyesus et al performed cross-sectional study about health care providers’ (HCPs) mental burden in six public hospitals in the south-west part of Ethiopia. They received responses from 322 participants, a response rate of 95%. They used Generalized Anxiety Disorder scale (GAD-7), Patients Health Questionnaire (PHQ-9), and 10-item Perceived Stress Scale (PSS) as rating scales for anxiety, depression, and psychological distress, respectively. In their participants, 36% were shown to have moderate or greater anxiety, 25% had moderate or greater depression, and 31% had moderate or greater menta distress. They showed multiple risk factors for psychiatric symptoms in HCPs: low income was a risk factor for anxiety and mental distress, and having a family member with COVID-19 was a risk factor for anxiety and depression. While these results presented in this manuscript are valuable in the field, there are some major concerns:

Major points:

1. The authors seem to be confusing “AOR” with relative risk. For example, page 22 lines 3-5, “Health care providers whose age >40 years old were about eight times [AOR=7.983; 95% CI (1.443-44.174)], more likely to develop anxiety than health care providers whose age 18-25 years old”, is it correct? If AOR means adjusted odds ratio (unfortunately the authors did not spell AOR out in the manuscript), “an odds ratio of 8” is not synonymous with “an 8-fold increase in risk”.

2. The authors should clearly describe the specific methods for collecting information of GAD-7, PHQ-9, and PSS from participants. For example, mailing paper-based questionnaires, SNS, telephone interview, etc.

3. Did the authors not obtain written consent? If so, please clearly state how the authors ensure that the participants could later withdraw their consent.

4. Did the authors have a validated Ethiopian version of the GAD-7, PHQ-9, and PSS? If the authors translated the English version of these questionnaires independently, did the authors train questioners to ask participants exactly the same way? Were there any biases in the answers given by different questioners?

5. The authors should add information on social restrictions in Ethiopia during the study period. Were strict social restrictions with legal penalties enforced during the study period?

Minor points:

1. Please correct the text in the manuscript: there are many inconsistent notations, unexplained abbreviations, and mistakes in spaces, dots, colons, and text color.

2. Is there any differences in the ratio of men to women by occupation?

3. Page 27 lines 10-12, “They are also more likely to suffer … limited access to accurate information and facts about the COVID-19 pandemic”, does it apply to skilled HCPs over the age of 40 as well?

4. Page 27 lines 16-19, this part looks confusing. It is natural for HCPs who already have infected family members to be concerned about the health of themselves and their affected family members. It should be distinguished from the HCPs feeling that they might infect their family members with COVID-19 in the future.

5. To describe income, please use the U.S. dollar or Euro as a key currency or use a simple classification such as “low income” or “high income”. Otherwise, it is difficult for a foreigner to imagine the relationship between local prices and income.

6. Page 28 lines 14-16, the authors should pick up the references with similar conditions to their study and add further discussion.

7. Page 28 lines 17-23, it looks confusing. Do the authors consider female HCPs to be home-makers? Female HCPs have the same social responsibilities as men as HCPs. Are there gender differences in occupations and income in the authors’ study, and do women tend to work in occupation with lower income and less responsibility?

8. There are duplicate sentences: page 27 line 14- and page 30 line 6-, page 27 line 20- and page 31 line 1-.

9. About figure 1, why is the sum of the numbers in the figure 325, but the inside of the oval is listed as 334?

10. About Figure 2 to 4, is there a reason why the figure should be in 3D?

11. About Figure 2, the total number in the figure is 322, which is different from the description in the text.

12. About Figure 4, what do Yes/No mean? What does the bar for the frequency mean?

13. About table 2, what do “husband” mean? If the authors mean “a married couple living together”, they should use "spouse”.

6. PLOS authors have the option to publish the peer review history of their article (what does this mean?). If published, this will include your full peer review and any attached files.

Reviewer #1: No

---

## [Author Response · Author response to Decision Letter 0]

14 Oct 2021

To: PLOS ONE

Subject: Replay to Review Report #1

Manuscript Ref. No: [PONE-D-21-20584] - [EMID:528479536e1a8b26]

Manuscript title: “Levels and predictors of anxiety, depression, and stress during COVID-19 pandemic among frontline healthcare providers in Gurage zonal public hospitals, Southwest Ethiopia, 2020: A Multicenter Cross-Sectional Study:”

Date: September 23/ 2021

Authors: 

Fisha Alebel GebreEyesus* 1, Tadesse Tsehay Tarekegn1 , Baye Tsegaye Amlak 1, Bisrat Zeleke Shiferaw1 , Mamo Solomon Emeria 1, Omega Tolessa Geleta 1 Tamene Fetene Terefe 1, Mtiku Mamo2, Melkamu Senbeta Jimma 3, Fatuma Seid Degu14, Elias Nigusu Abdisa5 , Menen Amare Eshetu6, Natnael Moges Misganaw7, Ermias Sisay Chanie 7 

Dear Reviewers and editors:

Greetings of the day!

First of all, we would like to present our gratitude acknowledgment, and appreciation for the effort you made to improve our manuscript throughout the review process during this difficult time of the COVID-19 pandemic. Also, we wish to thank you for considering this manuscript for publication in your journal.

Dear Reviewers and editors, on behalf of the authors I am submitting the revised version of the manuscript. We have gone through your constructive comments and question and devote all our effort to incorporate the feedback. All the authors are grateful to the reviewers and editors for their candid comments and timely communication. 

All the essential revisions are incorporated in the main manuscript and next to this cover letter please, kindly have a point-by-point guide on the response given to the comments/concerns and questions raised by reviewer # 1.

 Sincerely,

 Fisha Alebel GebreEyesus (MSc) - (corresponding author) on behalf of all authors)

Response to Reviewer#1

Dear reviewer,

First of all, we would like to present our gratitude acknowledgment, and appreciation for the effort you made to improve our manuscript throughout the review process during this difficult time of the COVID-19 pandemic. We also would like to express our heartfelt gratitude for your candid comments which we have addressed to the best of our abilities to improve the quality of our manuscript.

Reviewer comment/question # 1 Major points:

1. The authors seem to be confusing “AOR” with relative risk. For example, page 22 lines 3-5, “Health care providers whose age >40 years old were about eight times [AOR=7.983; 95% CI (1.443-44.174)], more likely to develop anxiety than health care providers whose age 18-25 years old”, is it correct? If AOR means adjusted odds ratio (unfortunately the authors did not spell AOR out in the manuscript), “an odds ratio of 8” is not synonymous with “an 8-fold increase in risk”. 

Response by the authors

Dear reviewer, 

Thank you for your constructive comments;

It is corrected as an Adjusted odds ratio based on your kind and constructive comment. Please kindly refer to the list of abbreviations in declaration section page 35 line number 6

Reviewer comment/question # 2 Major points:

2. The authors should clearly describe the specific methods for collecting information on GAD-7, PHQ-9, and PSS from participants. For example, mailing paper-based questionnaires, SNS, telephone interviews, etc.

Response by the authors

Dear reviewer, 

Thank you for your constructive comments;

It is stated and clearly explained under the data quality assurance and control section.

Data was collected from different healthcare workers in their respective wards using paper-based questionnaires. please kindly refer to page 12 line number 2-3

Reviewer comment/question # 3 Major points:

3. Did the authors not obtain written consent? If so, please clearly state how the authors ensure that the participants could later withdraw their consent 

Response by the authors

Dear reviewer, 

Thank you for your constructive comments 

As we all know Ethics is the branch of philosophy that deals with distinctions between

rights and wrongs. It is strongly recommended after Tuskegee Syphilis Study conducted on

the natural course of untreated syphilis among rural black males in Macon Country, Alabama

The purpose of ethical review is to consider the features of a proposed study in light of

ethical principles. To ensure that investigators have anticipated and satisfactorily resolved possible ethical objections and to assess their response to ethical issues raised by the study.

 The purpose of the study was explained to the study participants; anonymity, privacy, and confidentiality were ensured. As we know informed consent may be written or verbal consent. In our case, the health care providers are neither vulnerable to foreseeable risks and discomforts nor require compensation for possible injuries/harms due to participation in our studies so only verbal consent is necessary. So, the respondents’ right to refuse or withdraw from participating in the study was also fully acknowledged. We know whether the respondents were withdraw from the study in terms of response rate. If the response rate was 100% all eligible participants are fully engaged.

Reviewer comment/question # 4 Major points:

4. Did the authors have a validated Ethiopian version of the GAD-7, PHQ-9, and PSS? If the authors translated the English version of these questionnaires independently, did the authors train questioners to ask participants the same way? Were there any biases in the answers given by different questioners?

Response by the authors

Dear reviewer, 

Thank you for your constructive comments 

Yes, we have a validated Ethiopian version of the GAD-7, PHQ-9, and PSS. We measured symptoms of anxiety, depression, and stress using Generalized Anxiety Disorder 7-item (GAD-7), Patient Health Questionnaire 9-item (PHQ-9), and Perceived Stress Scale 10-item (PSS-10) respectively. The score of each measurement scale was used for anxiety symptoms (0 to 21), depression symptoms (0 to 27), and stress symptoms (0 to 40) in the analysis. The measurement item (GAD-7, PHQ-9, and PSS-10) is widely used in different research and validated in different settings and population groups. GAD-7, PHQ-9, and PSS-10 have been validated in Ethiopia using different cut-off points. 

To ensure the quality data, the questionnaire was translated from English into the local language (i.e., “Amharic”) for appropriateness and easiness in approaching the study participants and retranslated to English for analysis using language experts. . A two-day extensive training was given for both data collectors and supervisors before data collection, and common understanding was established on the data collection method and procedure. As a result of this, there was no bias rising from answering the questions raised by different data collectors. Moreover, the questionnaires were a four-point Likert scale ranging from 0 (“Not at all”) to 3 (“Nearly every day”) for anxiety and depression and a 5-point Likert-type scale ranging from 0 (never) to 4 (very often) for perceived stress which is clear, simple and didn’t result in any response bias and mutual agreement is reached between the data collector and the study participants in each value.

Reviewer comment/question # 5 Major points:

5. The authors should add information on social restrictions in Ethiopia during the study period. Were strict social restrictions with legal penalties enforced during the study period?

Response by the authors

Dear reviewer, 

Thank you for your constructive comments. Please kindly refer to page 6 line number 16-23

Ever since the official announcement of confirmed cases of COVID-19 in Ethiopia on 13 March, the Government has taken several preventive and control measures to contain the spread of the disease in the country. These include, but are not limited to, screening of incoming passengers

at international entry points, temporary flight suspensions to 30 countries, border closures, compulsory 14-day quarantine for arriving passengers, the establishment of isolation and quarantine centers for suspected and confirmed cases, laboratory diagnosis of COVID-19, procurement of medical supplies, community engagement, and risk awareness campaigns, resource mobilization, the activation of the Federal Emergency Operation Center, market control, disinfection of public transportation, temporary closure of schools and higher education institutions, establishing alternative working modalities for public servants, and the suspension of mass gatherings. Despite the ongoing preventative and control measures, containing the spread of the virus could be challenging in light of the underlying social and infrastructural settings of the country. Additionally, in Ethiopia, inherent cultural norms such as ritualistic greetings and strong social ties challenge key prevention methods such as social

distancing 

Reviewer comment/question # 1 Minor points

1. Please correct the text in the manuscript: there are many inconsistent notations, unexplained abbreviations, and mistakes in spaces, dots, colons, and text color

Response by the authors

Dear reviewer, 

Thank you for your constructive concern;

It is corrected based on your constructive comments.

Reviewer comment/question # 2 Minor points

2. Are there any differences in the ratio of men to women by occupation? 

Response by the authors

Dear reviewer, 

Thank you for your constructive comments 

Based on our findings from the total 322 participants enrolled in the study, the proportion of male health care providers was found to be 167(51.9%) whereas the number of female participants was 155 (48.1%). So, we can infer from this there was nearly equal involvement of male and female health providers in our study.

Reviewer comment/question # 3 Minor points

3. Page 27 lines 10-12, “They are also more likely to suffer … limited access to accurate information and facts about the COVID-19 pandemic”, does it apply to skilled HCPs over the age of 40 as well?

Response by the authors

Dear reviewer, 

Thank you for your constructive concern;

It is corrected by your constructive comment. Please kindly refer to page 29 line number 20-22, page 30 1-6 

Reviewer comment/question # 4 Minor points

4. Page 27 lines 16-19, this part looks confusing. It is natural for HCPs who already have infected family members to be concerned about the health of themselves and their affected family members. It should be distinguished from the HCPs feeling that they might infect their family members with COVID-19 in the future.

Response by the authors

Dear reviewer, 

Thank you for your constructive concern;

It is corrected based on the comments provided. Please kindly refer to page 30 lines 9-12

Reviewer comment/question # 5 Minor points

5. To describe income, please use the U.S. dollar or Euro as a key currency or use a simple classification such as “low income” or “high income”. Otherwise, it is difficult for a foreigner to imagine the relationship between local prices and income.

Response by the authors

Dear reviewer, 

Thank you for your constructive concern;

It is corrected as low income and high income based on your constructive comment. Please kindly refer to Tables 1, 4 and 8

Reviewer comment/question # 6Minor points

6. Page 28 lines 14-16, the authors should pick up the references with similar conditions to their study and add further discussion.

Response by the authors

Dear reviewer, 

Thank you for your constructive concern;

It is corrected based on your constructive comment. Please kindly refer to pages 31:5-9

Reviewer comment/question # 7Minor points

7. Page 28 lines 17-23, it looks confusing. Do the authors consider female HCPs to be home-makers? Female HCPs have the same social responsibilities as men as HCPs. Are there gender differences in occupations and income in the authors’ study, and do women tend to work in occupations with lower income and less responsibility?

Response by the authors

Dear reviewer, 

Thank you for your constructive concern;

It is corrected based on your comment. Please kindly refer to page 31 line number 11-19

Reviewer comment/question # 8Minor points

8. There are duplicate sentences: page 27 line 14- and page 30 line 6-, page 27 line 20- and page 31 line 1-.

Response by the authors

Dear reviewer, 

Thank you for your constructive concern;

It is corrected based on your comment. Please kindly refer to page 30: line 9-12, page 30 line 15-17, page 33 line 15-20

Reviewer comment/question # 9Minor points

9. About figure 1, why is the sum of the numbers in figure 325, but the inside of the oval is listed as 334?

Response by the authors

Dear reviewer, 

Thank you for your constructive concern;

Our total sample size was 334 but there were 322 study participants involved in the study with a response rate of 96.5%. So we correct it based on your comment. Please kindly refer to figure 1

Reviewer comment/question # 10 Minor points

10. About Figures 2 to 4, is there a reason why the figure should be in 3D?

Response by the authors

Dear reviewer, 

Thank you for your constructive concern;

We have used 3D to increase the resolution and quality of figures. Based on your comment we have changed figure 2 and figure 4 into 2D. please kindly refer to figure 2 and figure 4

Reviewer comment/question # 11 Minor points

11. About Figure 2, the total number in the figure is 322, which is different from the description in the text.

Response by the authors

Dear reviewer, 

Thank you for your constructive concern;

It is corrected based on your kind and constructive comment. 

Reviewer comment/question # 12 Minor points

12. About Figure 4, what do Yes/No mean? What does the bar for the frequency mean?

Response by the authors

Dear reviewer, 

Thank you for your constructive concern

 For the sake of clarity, we have changed the figure. Please kindly refer to figure 4

Yes Implies the magnitude of anxiety, depression, and stress Whereas No means the difference between 100% to the magnitude of each mental health problem respectively. 

The bar frequency on the right side shows the exact frequency of each mental health problem faced by health care providers. i.e 36%, 25.8%, and 31.4% of health care providers working in Gurage zonal Hospital had anxiety, depression, and stress respectively. Whereas the frequency mentioned on the left side indicates the proportion of health care providers who had no anxiety, depression, and stress respectively. 

Reviewer comment/question # 13 Minor points

13. About table 2, what does “husband” mean? If the authors mean “a married couple living together”, they should use "spouse”.

Response by the authors

Dear reviewer, 

Thank you for your constructive concern;

It is corrected as Spouse. Please kindly refer to table 2 page 15

 Yours sincerely,

 Fisha Alebel GebreEyesus (MSc) - (corresponding author) on behalf of all authors)

,

---

## [Decision Letter · Decision Letter 1]

19 Oct 2021

PONE-D-21-20584R1Levels and predictors of anxiety, depression, and stress during COVID-19 pandemic among frontline healthcare providers in Gurage zonal public hospitals, Southwest Ethiopia, 2020: A Multicenter Cross-Sectional StudyPLOS ONE

Dear Dr. GebreEyesus,

Thank you for submitting your manuscript to PLOS ONE. After careful consideration, we feel that it has merit but does not fully meet PLOS ONE’s publication criteria as it currently stands. Therefore, we invite you to submit a revised version of the manuscript that addresses the points raised during the review process.

We look forward to receiving your revised manuscript.

Kind regards,

Kensaku Kasuga

Academic Editor

PLOS ONE

Additional Editor Comments:

Please respond to the reviewer's comments.

Reviewers' comments:

Reviewer's Responses to Questions

**Comments to the Author**

1. If the authors have adequately addressed your comments raised in a previous round of review and you feel that this manuscript is now acceptable for publication, you may indicate that here to bypass the “Comments to the Author” section, enter your conflict of interest statement in the “Confidential to Editor” section, and submit your "Accept" recommendation.

Reviewer #1: (No Response)

2. Is the manuscript technically sound, and do the data support the conclusions?

Reviewer #1: Partly

3. Has the statistical analysis been performed appropriately and rigorously? 

Reviewer #1: Yes

4. Have the authors made all data underlying the findings in their manuscript fully available?

Reviewer #1: Yes

5. Is the manuscript presented in an intelligible fashion and written in standard English?

Reviewer #1: Yes

6. Review Comments to the Author

Reviewer #1: The authors have addressed the most of the issues except for the following.

They still mistakes “odds ratio” for “relative risk”. “Odds ratio = 2.0” dose not mean “two times more likely”. We can say “two times more likely” only when the relative risk is 2.0 in a prospective cohort study. As this study is a cross-sectional case-control studies, they cannot mention the relative risks. The odds ratio can be occasionally used as an approximation of the relative risk when patients with diseases are less than 1% in general. However, the odds ratio often dissociates from the relative ratio in cohorts including many patients with diseases. Considering the substantial numbers of participants with symptoms in the study, they should not handle the odds ratio as the relative risk.

For example (in the result section):

“Health care providers whose age >40 years old were about eight times

[AOR=7.983; 95% CI (1.443-44.174)], more likely to develop anxiety than health care

providers whose age 18-25 years old”,

I agree that the difference is statistically significant, but the risk is not “eight times”.

So, how about just changing like the below?

“Health care providers whose age >40 years old were significantly more likely to develop anxiety than health care providers whose age 18-25 years old” [AOR=7.983; 95% CI (1.443-44.174)],

Please also revise the many other descriptions regarding AOR.

7. PLOS authors have the option to publish the peer review history of their article (what does this mean?). If published, this will include your full peer review and any attached files.

Reviewer #1: No

---

## [Author Response · Author response to Decision Letter 1]

26 Oct 2021

To: PLOS ONE

Subject: Replay to Review Report #2

Manuscript Ref. No: [PONE-D-21-20584] - [EMID:528479536e1a8b26]

Manuscript title: “Levels and predictors of anxiety, depression, and stress during COVID-19 pandemic among frontline healthcare providers in Gurage zonal public hospitals, Southwest Ethiopia, 2020: A Multicenter Cross-Sectional Study:”

Date: October 26/ 2021

Authors: 

Fisha Alebel GebreEyesus* 1, Tadesse Tsehay Tarekegn1 , Baye Tsegaye Amlak 1, Bisrat Zeleke Shiferaw1 , Mamo Solomon Emeria 1, Omega Tolessa Geleta 1 Tamene Fetene Terefe 1, Mtiku Mamo2, Melkamu Senbeta Jimma 3, Fatuma Seid Degu14, Elias Nigusu Abdisa5 , Menen Amare Eshetu6, Natnael Moges Misganaw7, Ermias Sisay Chanie 7 

Dear Reviewers and editors:

Greetings of the day!

First of all, we would like to present our gratitude acknowledgment, and appreciation for the effort you made to improve our manuscript throughout the review process during this difficult time of the COVID-19 pandemic. Also, we wish to thank you for considering this manuscript for publication in your journal.

Dear Reviewers and editors, on behalf of the authors I am submitting the revised version of the manuscript. We have gone through your constructive comments and question and devote all our effort to incorporate the feedback. All the authors are grateful to the reviewers and editors for their candid comments and timely communication. 

All the essential revisions are incorporated in the main manuscript and next to this cover letter please, kindly have a point-by-point guide on the response given to the comments/concerns and questions raised by reviewer # 1.

 Sincerely,

 Fisha Alebel GebreEyesus (MSc) - (corresponding author) on behalf of all authors)

Response to Reviewer#1

Dear reviewer,

First of all, we would like to present our gratitude acknowledgment, and appreciation for the effort you made to improve our manuscript throughout the review process during this difficult time of the COVID-19 pandemic. We also would like to express our heartfelt gratitude for your candid comments which we have addressed to the best of our abilities to improve the quality of our manuscript.

 Review Comments to the Author

Reviewer #1: The authors have addressed the most of the issues except for the following.

They still mistakes “odds ratio” for “relative risk”. “Odds ratio = 2.0” dose not mean “two times more likely”. We can say “two times more likely” only when the relative risk is 2.0 in a prospective cohort study. As this study is a cross-sectional case-control studies, they cannot mention the relative risks. The odds ratio can be occasionally used as an approximation of the relative risk when patients with diseases are less than 1% in general. However, the odds ratio often dissociates from the relative ratio in cohorts including many patients with diseases. Considering the substantial numbers of participants with symptoms in the study, they should not handle the odds ratio as the relative risk.

For example (in the result section): “Health care providers whose age >40 years old were about eight times [AOR=7.983; 95% CI (1.443-44.174)], more likely to develop anxiety than health care providers whose age 18-25 years old”, 

I agree that the difference is statistically significant, but the risk is not “eight times”. So, how about just changing like the below? “Health care providers whose age >40 years old were significantly more likely to develop anxiety than health care providers whose age 18-25 years old” [AOR=7.983; 95% CI (1.443-44.174)], Please also revise the many other descriptions regarding AOR.

Response by the authors

Dear reviewer, 

Thank you for your constructive comments;

It is corrected based on your kind and constructive comment. Please kindly refer to page 19 line number 15-19, page 20 line 1-5, page 23 line 11-20, page 24 line 1-2 and page 27 line 13-18

2. Is the manuscript technically sound, and do the data support the conclusions?

Reviewer #1: Partly

Response by the authors

Dear reviewer, 

Thank you for your constructive comments;

It is corrected based on the comment. Please kindly refer page 34 line 16-21 and page 35 line 1-2

 Yours sincerely,

 Fisha Alebel GebreEyesus (MSc) - (corresponding author) on behalf of all authors)

---

## [Decision Letter · Decision Letter 2]

29 Oct 2021

Levels and predictors of anxiety, depression, and stress during COVID-19 pandemic among frontline healthcare providers in Gurage zonal public hospitals, Southwest Ethiopia, 2020: A Multicenter Cross-Sectional Study

PONE-D-21-20584R2

Dear Dr. GebreEyesus,

We’re pleased to inform you that your manuscript has been judged scientifically suitable for publication and will be formally accepted for publication once it meets all outstanding technical requirements.

Kind regards,

Kensaku Kasuga

Academic Editor

PLOS ONE

Additional Editor Comments (optional):

All comments have been addressed

Reviewers' comments:

Reviewer's Responses to Questions

**Comments to the Author**

1. If the authors have adequately addressed your comments raised in a previous round of review and you feel that this manuscript is now acceptable for publication, you may indicate that here to bypass the “Comments to the Author” section, enter your conflict of interest statement in the “Confidential to Editor” section, and submit your "Accept" recommendation.

Reviewer #1: All comments have been addressed

2. Is the manuscript technically sound, and do the data support the conclusions?

Reviewer #1: Yes

3. Has the statistical analysis been performed appropriately and rigorously? 

Reviewer #1: Yes

4. Have the authors made all data underlying the findings in their manuscript fully available?

Reviewer #1: Yes

5. Is the manuscript presented in an intelligible fashion and written in standard English?

Reviewer #1: Yes

6. Review Comments to the Author

Reviewer #1: Thank you for revising the manuscript. All issues that I pointed out have been addressed now. This article is now acceptable for the journal.

7. PLOS authors have the option to publish the peer review history of their article (what does this mean?). If published, this will include your full peer review and any attached files.

Reviewer #1: No

---

## [Editor Report · Acceptance letter]

4 Nov 2021

PONE-D-21-20584R2 

Levels and predictors of anxiety, depression, and stress during COVID-19 pandemic among frontline healthcare providers in Gurage zonal public hospitals, Southwest Ethiopia, 2020: A Multicenter Cross-Sectional Study 

Dear Dr. GebreEyesus:

I'm pleased to inform you that your manuscript has been deemed suitable for publication in PLOS ONE. Congratulations! Your manuscript is now with our production department. 

Kind regards, 

on behalf of

Dr. Kensaku Kasuga 

Academic Editor

PLOS ONE